

# A semi-empirical error estimation for PWV derived from atmospheric radiosonde data at the Canary Islands.

Julio A. Castro-Almazán[1,2], Gabriel Pérez-Jordán[1,*], and Casiana Muñoz-Tuñón[1,2]

[1]Instituto de Astrofísica de Canarias, E-38200, La Laguna, Spain
[2]Dept. Astrofísica, Universidad de La Laguna, E-38200, La Laguna, Spain
[*]Visiting fellow.

*Correspondence to:* JACA (jcastro@iac.es); GPJ (gpj@iac.es); CMT (cmt@iac.es)

**Abstract.** An unbiased method to estimate the error and the optimum number of sampled levels in Precipitable Water Vapour (PWV) determinations from atmospheric radiosoundings is proposed. Two components have been considered, the uncertainties in the measures and the sampling error. The sampling component has been modelled from an empirical dataset of 64 high vertical resolution

radiosounding profiles equipped with sondes Vaisala RS80 and RS92. The balloons were launched at the astronomical Roque de los Muchachos Observatory (ORM, $\sim 2200$ masl), during intensive and unique site testing runs carried out in 1990 and 1995, and from the neighbour operational station of Güímar, in Tenerife (TFE, $\sim 105$ masl) in $2013-2014$. The PWV values ranged between $\sim 0.9$ mm and $\sim 41$ mm. The method takes into account the dependence on the number of samples measured,

after sub-sampling the profile for error minimization, and was tested by comparison with a dataset of 42 extremely low resolution profiles only sampling the standard levels ($\sim 15$ levels).

The results show that errors are larger for the wettest atmosphere conditions. On the other hand, drier conditions requires a larger optimum number of samples. The optimum number of samples $N_0$ is less than 200 for PWV $\gtrsim 10$ mm. For drier conditions, as in astronomical sites, $N_0$ grows up to $\sim$

550 levels. This result may be important for PWV determinations in astronomical observatories. The absolute errors are always $< 0.6$ mm, with a median relative error of $2.4 \pm 0.8\%$ and extreme value of $7.9\%$ in the driest condition (PWV$= 0.89$ mm). These errors reduce the uncertainties previously reported in the literature. Nevertheless, errors grow up to $30\%$ in poorly sampled profiles (the number of samples being less than $N_0$) for dry atmospheres.

Alternative equations for direct error estimation, specifically for PWV from radiosoundings equipped with Vaisala RS80 and RS92 sensors, are also provided.





## 1 Introduction

Water Vapour (WV) accounts only for a 0-4% of all atmospheric molecules. Nevertheless, it is a powerful Greenhouse Gas, with strong lines of absorption and emission in the Infrared (IR). Atmospheric
WV also participates in processes affecting the global climate (Elliott and Gaffen, 1995; Ahrens, 2003) and is the principal molecule responsible for the atmospheric extinction in IR astronomical observations, specially at wavelengths longer than $\sim$15 microns (Far IR), in several bands in the Mid and Short Wavelength IR and also in the submillimeter and microwave range (Selby and Mampaso, 1991; Hammersley, 1998; García-Lorenzo et al., 2010; Otárola et al., 2010).

The total amount of WV above a particular location is highly variable and can be expressed as the Precipitable Water Vapour (PWV), that is defined as the total water column height if integrated from the surface to the top of the atmosphere with unit cross-sectional[1]. PWV is commonly expressed in mm, in terms of the height to which that water would stand if completely condensed and collected in a vessel with a cross section of $1$ m$^2$.

PWV can be measured by equipped radiosounding balloons, radiometers both from ground (Fowle, 1912; Guiraud et al., 1979; Carilli and Holdaway, 1999; Smith et al., 2001) or satellites (Grody et al., 1980; Menzel et al., 1998; Gao and Kaufman, 2003; Deeter, 2007; Wong et al., 2015), sun photometers (Bird and Hulstrom, 1982; Volz, 1983; Plana-Fattori et al., 1998; Firsov et al., 2013), lunar photometers (Barreto et al., 2013), GPS receivers (Bevis et al., 1992, 1994), Fourier Transform Infrared
spectrometers (Kurylo, 1991; Schneider et al., 2006) and others (Schneider et al., 2010). Among them, atmospheric radiosoundings are a direct *in situ* measurement and one of the most accurate techniques to retrieve the PWV. Radiosoundings are also one of the current standards for atmospheric research and are widely used as a valid reference for comparisons or calibrations. An error estimation of PWV from radiosoundings is therefore extremely important. Previous works have dealt
with the issue of accuracy of radiosonde measurements of PWV including experimental errors, differences in sensors output because variations of wetness in the column or comparison of different sensor types, see for example Miloshevich et al. (2006) or Romero-Campos et al. (2011). Nevertheless, as radiosondes provide a discrete profiling of the atmosphere column, other important factors impacting the final error in PWV measurements are the error propagation and the vertical sampling
(Liu et al., 2000). These last error sources have not been so much explored and are the issue of the present work.

### 1.1 Rationale and objectives

An accurate error estimation of PWV from radiosoundings is essential while making regression analyses in comparisons or calibration studies. In this sense, special care is needed when radiosoundings
from different sources and characteristics are being included or when working in dry environments.

---

[1]http://glossary.ametsoc.org/wiki/Precipitable_water





This is the case in Pérez-Jordán et al. (2015), where we used a set of 23 high vertical resolution balloons to validate the Weather Research and Forecasting mesoscale numerical model (WRF) for a dry astronomical location (Roque de los Muchachos Observatory, ORM hereafter; see Sect. 1.2 for a detailed description of all the locations and datasets). The possibility of making use of these radiosounding datasets supposed an opportunity, as no other atmospheric balloons have been launched at ORM and they provide a very high vertical resolution dataset with 2500 to 7000 datapoints per balloon flight.

A second sample of 42 radiosoundings were also included in Pérez-Jordán et al. (2015) for verification and control. The point selected was the radiosounding operational station located close to the sea level in the neighbour island of Tenerife (TFE, hereafter). The data were downloaded from the repository at the Department of Atmospheric Science of the University of Wyoming[2] (WYO hereafter). Both datasets, ORM and TFE-WYO, showed strong differences in sampling ($\sim 100$ levels at TFE-WYO versus more than 2500 at ORM) as a consequence of the re-encoding applied at WYO.

In the present work, we took the opportunity of using the data from the high vertical resolution radiosounding flights at ORM to model the error in PWV calculations from balloons and suggest a method to retrieve it. To extend the validity of the model, we have added the dataset of 42 radiosoundings at TFE described in Pérez-Jordán et al. (2015). In this work we obtained the data from the Spanish State Meteorological Agency[3] (AEMET, following the spanish acronym) to include all the available operational datapoints ($> 2500$). The model covers a wide variety of conditions originated by the different path lengths (departures from $105$ masl and $\sim 2200$ masl) and PWV content ($\sim 10 - 40$ mm at TFE and $\sim 1 - 13$ mm at ORM).

The method takes into account the propagation of uncertainty and the sampling error. The result also includes an estimation of the optimum number of sampled levels needed to fit into the minimum error as a function of the expected PWV content. To validate the model, we have selected only the standard operational levels from the TFE dataset ($\sim 15$ datapoints each radiosounding), applied the method and compared with the full dataset.

## 1.2 Locations and datasets

The balloons were launched at two separated locations (see Fig. 1): ORM at the summit of La Palma Island and the World Meteorological Organization (WMO) station 60018[4] at Güímar, Tenerife Island (TFE), both in the Canary Islands. ORM is one of the most competitive astronomical observatories in the world, hosting, among others, the current largest optical-IR telescope, the $10.4$ m Gran Telescopio Canarias (GTC)[5]. The altitude at ORM ranges from $\sim 2200$ masl to $\sim 2400$ masl, being the launching point of the balloons in the lowest level. TFE station is the closest radiosounding opera-

---

[2]http://weather.uwyo.edu/upperair/sounding.html
[3]http://www.aemet.es/en/
[4]http://weather.gladstonefamily.net/site/60018
[5]http://www.gtc.iac.es



tional station, located 155 km away and close to the sea level (105 masl). The balloons at TFE are routinely launched twice a day (00 and 12 UTC) by the AEMET.

Because of their latitude, the strong influence from the Azores high and the cold oceanic stream, the lower troposphere of the Canary Islands exhibits a vertical structure with an almost constant thermal inversion layer (IL). The altitude of the IL ranges on average from $800$ m in summer to $1600$ m in winter, well below the altitude of the ORM (Dorta-Antequera, 1996). The IL separates the moist marine boundary layer and the dry free atmosphere, inducing very high atmospheric stability above it. Therefore, the PWV condition at both locations, TFE and ORM, are strongly uncorrelated and show wide differences in atmospheric conditions. The average mesoscale conditions in the Canary Islands show a relative humidity profile drastically drooping with altitude. Temperatures below $-40°$C are typically reached above $8$ km in the troposphere.

The ORM data comes from a very unique dataset obtained during intensive site testing assessments performed in April 1990, July 1990 and November 1995 in a join run carried out by the Instituto de Astrofísica de Canarias and the University of Nice with support of the Centre National de Recherches Météorologiques (Météo-France). The details of these campaigns are described in Vernin and Muñoz-Tuñón (1992, 1994). A total of 23 balloons were launched from the Observatory, 22 of them were selected to model the error, after rejecting the one that did not ranged the whole troposphere. The data from TFE covers one year of data with a total of 42 operational soundings from May 2013 to April 2014. Therefore, the join dataset used in the model includes 64 radiosounding with more than 2500 vertical levels each ($\sim 2$ s of time resolution) and PWV contents ranging between $< 1$ mm and $> 40$ mm.

The radiosondes used were Vaisala RS80 at ORM (1990 and 1995) and Vaisala RS92 for the operational soundings at TFE (2013-2014). In both cases, the sondes were calibrated in the ground check immediately before the balloon release. The Vaisala standard procedure for sonde preparation and sistematic errors sources minimization were followed, besides the different corrections for day time solar heatings, etc. For an extensive characterization of both radiosonde set-ups see Miloshevich et al. (2001, 2009). In the particular case of TFE, more details on the corrections applied are in Romero-Campos et al. (2011).

The uncertainties assumed in this work for both radiosondes are $\pm 0.5°$C, $\pm 5\%$ and $\pm 1$ hPa for temperature, relative humidity and atmospheric pressure. These values are in agreement with the estimates for the RS92 obtained within the extensive network of the GCOS (Global Climate Observing System) Reference Upper-Air Network[6] (GRUAN) by Dirksen et al. (2014). Future improvements on sensors uncertainties can be easily implemented, as for example, the inclusion of vertically resolved profiles of uncertainties (Dirksen et al., 2014) that the new binary encodings for radiosonde data, as BUFR (Dragosavac, 2007) are introducing.

---

[6]http://www.gruan.org





## 2 PWV from radiosondes

The PWV is obtained from the temperature $T$ (°C), atmospheric pressure $p$ (hPa) and relative humidity RH (%) measured by the radiosondes (Curry and Webster, 1999). Following the definition given in Sect. 1, the PWV can be expressed by:

$$\text{PWV} = \frac{10^3}{\rho} \int\limits_{z=0}^{z=\infty} \rho_w dz \qquad \text{(mm)}, \tag{1}$$

where $z$ is the height in m and $\rho$ and $\rho_w$ are the liquid water and WV densities, both in $\text{kg/m}^3$. The definition of WV mixing ratio $r$ is

$$r = \frac{m_w}{m_d} = \frac{\rho_w}{\rho_d}, \tag{2}$$

where $m_w$ and $\rho_w$ are the mass and density of WV and $m_d$ and $\rho_d$ are the corresponding values for dry air. We can assume hydrostatic balance ($dp = -\rho_d \cdot g \cdot dz$) and write Eq. 1 in the form

$$\text{PWV} = \frac{10^5}{\rho g} \int\limits_{p_t}^{p_s} r dp \qquad \text{(mm)}, \tag{3}$$

where $g$ is the Earth gravity ($\text{m/s}^2$) and $p_s$ and $p_t$ are the pressure at surface and top of the sampled atmospheric column in hPa. We can now apply the ideal gas law and the Dalton's law of partial pressures to Eq. 2 resulting,

$$r = 0.622 \left( \frac{e}{p-e} \right), \tag{4}$$

where the coefficient 0.622 is the molecular mass ratio of WV in dry air and $e$ is the partial vapour pressure that can be obtained from the definition of relative humidity as

$$e = e_{sat} \cdot \frac{\text{RH}}{100} \qquad \text{(hPa)}, \tag{5}$$

where the saturation vapour pressure $e_{sat}$ can be expressed as an empirical polynomial fit, following Curry and Webster (1999):

$$e_{sat} = a_0 + T \left( a_1 + T \left( a_2 + T \left( a_3 + T \left( a_4 + T \left( a_5 + T a_6 \right) \right) \right) \right) \right) \qquad \text{(hPa)}. \tag{6}$$

The coefficients $a_i$ have been taken from Flatau et al. (1992) (see Table 1).

## 3 Error budget for PWV from radiosonde data

The integral in Eq. 3 is computed as a discrete summation over all the levels sampled by the radiosonde after a trapezoidal method. In this sense, the error will have two components, one associated to the uncertainty of the measure ($\sigma$) and other to the sampling ($\epsilon_s$). Let $\epsilon_f$ be the final error associated to the PWV determination from radiosondes, then we can separate $\epsilon_f$ in:

$$\epsilon_f^2 = \sigma^2 + \epsilon_s^2. \tag{7}$$



### 3.1 Uncertainty propagation

Let $\sigma_p$, $\sigma_T$ and $\sigma_{RH}$ be the instrumental uncertainties associated to the direct measure of atmospheric pressure $p$, temperature $T$ and relative humidity RH. The uncertainty of PWV can be obtained by error propagation over all the sampled levels $N$. Applying the chain rule to the trapezoidal method in Eq. 3 leads to:

$$\sigma^2 = \left(\frac{10^5}{\rho g}\right)^2 \sum_{i=0}^{N-1} \left[\frac{1}{4}(\Delta p_i)^2 \left(\sigma_{r,i+1}^2 + \sigma_{r,i}^2\right) + 2 \cdot \sigma_p^2 \cdot r_{i,avg}^2\right] \quad \text{(mm)}, \tag{8}$$

where $p_i$ is the pressure at level $i$, $\Delta p_i = p_{i+1} - p_i$, $r_{i,avg} = (r_{i+1} + r_i)/2$ and $r_i$ is the mixing ratio at level $i$. Replace $(2 \cdot \sigma_p^2)$ by $(\sigma_{p,i+1}^2 + \sigma_{p,i}^2)$ for vertically resolved sensor uncertainties. By a recursive use of the error propagation rules in Eqs. 4, 5 and 6, the following expressions are obtained for the uncertainties of water vapour mixing ratio $\sigma_{r,i}$, partial vapour pressure $\sigma_{e,i}$ and saturation vapour pressure $\sigma_{esat,i}$, all at level $i$.

$$\sigma_{r,i}^2 \approx \left(0.622 \cdot \frac{\sigma_{e,i}}{p_i}\right)^2, \tag{9}$$

$$\sigma_{e,i}^2 = 10^{-4} \cdot \left[(RH_i \cdot \sigma_{esat,i})^2 + (e_{sat,i} \cdot \sigma_{RH})^2\right] \quad \text{(hPa)}, \tag{10}$$

$$\sigma_{esat,i}^2 = \sum_{j=0}^{6} \left[\left(T_i^j \cdot \sigma_{a,j}\right)^2 + \left(a_j \cdot j \cdot T_i^{j-1} \cdot \sigma_T\right)^2\right] \quad \text{(hPa)}, \tag{11}$$

where $a_j$ and $\sigma_{a,j}$ are the saturation vapour pressure coefficients and the associated uncertainties (see Table 1; use $\sigma_{RH,i}$ and $\sigma_{T,i}$ for vertically resolved sensor uncertainties). The contribution of the covariances in the propagation of errors was specifically calculated leading to negligible values, less than $10^{-7}$ mm and $10^{-3}$ mm, for ORM and TFE, respectively, and therefore, not included in the Eqs. 8 to 11.

### 3.2 Sampling error

The number of levels included for PWV determinations from atmospheric radiosondes may range from tens (when only standard levels are available) to thousands (full profile). The 64 balloons considered in this work range from $\sim 2500$ to $\sim 7000$ data points, dense enough to neglect the sampling error. This circumstance allowed to empirically approximate an analytical expression for $\epsilon_s$ as a function of the number of sampled levels $N$, following a recursive sub-sampling process.

Each profile was uniformly sub-sampled at equal intervals by taking one point in two, one in three, etc, up to obtain 800 different realizations of PWV for the same profile (any other uniform sub-sampling is valid). The dispersion of the residuals increases in a logarithm fashion as the number of levels $N$ decreases (see Fig. 2a), with the residuals defined as,

$$\text{res}_N = I - \widetilde{I}_N, \tag{12}$$





where $I$ and $\widetilde{I}_N$ are the integral in eq 3 calculated with all the levels in the profile ($N_{max}$) and with
the different realizations of $N$ levels after the sub-sampling process. The residuals were grouped
in slices for intervals of $N$ to fit in a model. The size of the slices was selected following a quasi-
logarithm scale to overcome the differences in variance (heteroscedasticity) while conserving the
statistical significance (see Fig. 2a for details on the slices and the number of residuals included for
each). The sample error was then obtained as the RMSE of the residuals for each slice. The RMSE
was calculated as the square root of the sum of the variance and the squared bias for every interval.

The dependence on $N$ of $\mathrm{res}_N$ come from the integrals $\widetilde{I}_N$ in Eq. 12 in the form,

$$\widetilde{I}_N = \widetilde{\mathrm{PWV}}(N^{-1}) + E(N^{-2}), \tag{13}$$

where the first term, $\widetilde{\mathrm{PWV}}$, is the composite trapezoidal sum of Eq. 3 sub-sampled to N levels and
the second one is the associated error. Therefore, taking $N$ as the middle point of each slice interval
in Fig. 2a we modelled the residuals (Eq. 12) behaviour by fitting a function $A/N + B/N^2 + C$
to the RMSEs, with the coefficient $C = 0$, as $\lim_{(N \to \infty)} \mathrm{res}_N = 0$. Finally, we applied a gradient-
expansion algorithm to compute a non-linear least squares fit, obtaining the following equation (red
line in Fig. 2).

$$\epsilon_s = \frac{30}{N} + \frac{234}{N^2} ; \quad (N \gtrsim 10). \tag{14}$$

This equation assumes that the radiosounding is uniformly sampling the whole atmosphere where
the PWV concentrates (mainly the lower and mid troposphere).

## 4   Optimized error

The integrated PWV is a summation and then, the uncertainty propagates with the contribution of
all the addends. Therefore, $\sigma$ increases with the number of levels in the profile. On the other hand,
the sampling error is on the opposite direction and $\epsilon_s$ increases as the number of sampled levels
decreases. This behaviour is described in Fig. 2b, which shows the sampling error $\epsilon_s$ fitted in Eq.
14 and the median $\sigma$ calculated as a function of the sampling levels $N$ for each slice of sub-sampled
data.

Therefore, it is always possible to uniformly sub-sample the profile, obtaining $\epsilon_f(N, \mathrm{PWV})$ by
use of Eq. 7 for different values of $N$ ($N \leq N_{max}$). Hence, the optimized error $\epsilon$ will result in
minimizing $\epsilon_f$ whilst reducing $N$ (see Fig. 2b).

$$\begin{aligned} \epsilon^2 &= \min[\epsilon_f^2(N, \mathrm{PWV})] \\ &= \min[\sigma^2(N, \mathrm{PWV}) + \epsilon_s^2(N)], \end{aligned} \tag{15}$$

where $\epsilon_f$ is the final error defined in Eq. 7, $\sigma$ is the propagated uncertainty (Eq. 8) and $\epsilon_s$ is the
sampling error (Eq. 14). The optimum number of samples $N_0$ will be defined as the argument of the





minimum in Eq. 15.

$$N_0 = \arg\min[\epsilon_f]. \tag{16}$$

Finally, we can calculate the individual contribution of the uncertainty and the sampling error to the optimized error by means of,

$$\sigma_0 = \sigma(N_0, \text{PWV}),$$

$$\epsilon_{s0} = \epsilon_s(N_0). \tag{17}$$

## 5 Results

We have applied the optimization described in Sect. 4 to all the available radiosondes (both ORM and TFE), recursively computing Eq. 7 while sub-sampling the profile and carrying out the minimization in Eq. 15. Fig. 3 shows six particular examples, two from ORM (Fig. 3a,b) and two from TFE (Fig. 3c,d), with different PWV concentrations that contain the extremes. Additionally, we have included the same two profiles showed for TFE ($N_{max} \gg N_0$) with the re-encoded sampled data downloaded from the WYO repository (see section 1.1), where $N_{max} \approx N_0$ (Fig. 3e,f). In all the cases, the residuals oscillate inside the range defined by the sampling error model (red line in Fig. 3) while converging. For $N_{max} \gg N_0$, the error is being dominated by $\sigma$, whereas for $N_{max} \approx N_0$ the sample error $\epsilon_s$ becomes significant. We sub-sampled the data in the profiles from WYO by extracting $N$, $N-2$, $N-4$, ... points, uniformly distributed, for each iteration, to obtain a sufficient number of realizations close to $N_{max}$.

Fig. 4 shows all the results. The parameters $\epsilon$ and $N_0$ are plotted together for all the available radiosondes (ORM and TFE) as a function of PWV. The optimized error $\epsilon$ is proportional to PWV whereas $N_0$ is the reverse. Therefore, lower amounts of PWV require more richly sampled profiles for an accurate estimation. The optimum sampling number $N_0$ is less than 200 for PWV $\gtrsim 10$ mm (see Fig. 4) while, for drier conditions, as in astronomical sites, $N_0$ grows up to $\gtrsim 550$ levels. All the numerical results are listed as an appendix in Tables 3 to 5.

We obtained errors less than $\sim 0.6$ mm for all the cases, that is significantly lower than the $1.3$ mm error published by Liu et al. (2000), also taking into account the sampling effect in the error, but with no deeper analyses onto the dependence on $N$. Specifically for the sampling component $\epsilon_{s0}$, we also obtained a lower average value of $0.2$ mm, with a maximum of $0.34$ mm, in comparison with $0.50$ mm and $\approx 1$ mm, respectively, obtained by Liu et al. (2000) after analysing the residuals between the smoothed data (the standard output) and the much denser real-time records from 50 PWV radiosoundings at Hong Kong.

The best fits to the data are also included in Fig. 4:

$$N_0 = (615 \pm 45) \cdot \text{PWV}^{(-0.51 \pm 0.01)}; \quad (RMSE = 31), \tag{18}$$





$$250 \quad \epsilon = (0.09 \pm 0.01) \cdot \mathrm{PWV}^{(0.53 \pm 0.01)} \qquad (\mathrm{mm}); \quad (RMSE = 0.02\,\mathrm{mm}). \qquad (19)$$

Equation 18 may be used for an *a priori* estimation of the optimum number of levels $N_0$ to be sampled by the radiosonde for measuring certain amount of PWV within the minimum error $\epsilon$, while Eq. 19 gives a direct estimation of such an error if $N_{max} \geq N_0$. Therefore, these equations may be used instead of the more time-consuming sub-sampling minimization of Sect. 4. Nevertheless, Eqs.
18 and 19 are estimators of $\epsilon$ and $N_0$ only for radiosondes with Vaisala RS80 or RS92 sondes with the instrumental uncertainties described in Sect. 1.2. Other experimental set-ups will require new curves to be fitted after propagating the uncertainties (see Sect. 3.1) and applying Eq. 15.

Figure 5 and Table 2 summarize the statistical results for the relative errors ($\epsilon_{rel}$ hereafter) from the whole dataset of 64 radiosoundings (ORM + TFE). Relative errors behave opposite than absolute
optimized errors $\epsilon$, in the sense that the drier the atmosphere, the larger the $\epsilon_{rel}$ (see Fig. 5a). The median error is $2.4 \pm 0.8\%$, with ninetieth percentile $P_{90} = 5.0\%$ and an extreme value of $7.9\%$ for the driest condition (PWV $= 0.89$ mm). The complete list of $\epsilon_{rel}$ is also included in the appendix, (Tables 3 to 5). These results reduce in more than a half the uncertainty of $\approx 5\%$ ($\approx 15\%$ for extremely drier conditions) published by Schneider et al. (2010) following the mixing ratio uncertainties obtained by Miloshevich et al. (2009) from the vaisala RS92 RH sensors, and bring out the importance
of an optimized sampling in the PWV determinations from radiosoundings.

### 5.1 Validation with poorly sampled radiosonde data ($N_{max} < N_0$)

To validate the model, we have selected only the standard operational levels from the TFE dataset (TFE-STD, hereafter) and compared with the high resolution profiles (see Tables 4 and 5), used
as the reference. The $\sim 15$ standard levels are the minima and mandatory levels reported by the radiosondes, but it constitutes a poorly sampled scenario ($N_{max} < N_0$). In this case, Eq. 19 underestimates the error and the sampling component dominates. Therefore, $\epsilon \approx \epsilon_s$ and the error can be directly estimated from Eq. 14 with $N = N_{max}$.

The results of the comparison[7] are shown in the Fig. 6, where both series are plotted with the
calculated errors. The high resolution PWV series falls inside the TFE-STD error margin and, therefore, the modelled errors statistically represent the differences between the calculated PWV and the best available estimation.

The relative errors for the TFE-STD series range between $7.1\%$ and $33.6\%$, with a median of $\sim 17.7\%$. Fig. 6 also evidences the ability of the standard levels to reproduce the tendencies in long term
PWV monitoring programs. A poorly sampled scenario may be also found when estimating the PWV from low resolution operational radiosoundings in dry atmospheres as, for example, astronomical observatories, where the balloons are usually launched below the observatory height and thence, the lowest levels must be trimmed.

---

[7]The flight 13335A has been removed from the TFE-STD series because the first standard level is missed.



## 6   Conclusions

We have considered two components in the error estimation for PWV obtained from radiosoundings, the uncertainties in the measures $\sigma$ and the sampling error $\epsilon_s$ (the whole atmospheric column must be uniformly sampled). The uncertainty contribution (Eq. 8) has been estimated by means of analytical error propagation through all the levels sampled by the balloon. The sampling component has been modelled from an empirical dataset of 64 high vertical resolution radiosounding profiles (Eq. 14) with PWV values ranging between $\sim 0.9\,\mathrm{mm}$ and $\sim 40\,\mathrm{mm}$. The model is based on the dependence on the number of samples of the composite trapezoidal formula for numerical integration.

The uncertainty $\sigma$ increases and $\epsilon_s$ decreases as the number of sampled levels grow. Therefore, we have optimized the error whilst reducing the number of samples $N$. The optimization (Eq. 15) leads to the calculation of the minimum error and may be considered an appropriate estimator of the final error in the PWV determination. The value of $N$ in the minimum error is the optimum number of samples $N_0$ (Eq. 16).

The sub-sampling minimization was applied to all the radiosounding datasets. The results show a coherent behaviour with no differences or bias as a function of the profile. The largest errors were found for the wettest atmosphere conditions. On the other hand, the drier the conditions, the larger the optimum number of samples $N_0$ needed to fit into the minimum error. The value of $N_0$ is less than 200 for PWV $\gtrsim 10\,\mathrm{mm}$ (see Fig. 4). For drier conditions, as in astronomical sites, $N_0$ grows up to $> 550$ levels. The absolute errors are always $< 0.6\,\mathrm{mm}$, with the sampling component $\epsilon_{s0} < 0.35$ mm.

Two different scenarios arise after the determination of $N_0$, whether the actual number of levels in the profile $N_{max}$ is greater or equal than $N_0$ or not. For $N_{max} \geq N_0$ is always possible to reach $N_0$ by sub-sampling and, therefore, the minimum error. The error behaviour with the PWV value, in this case, is shown by the Eqs. 18 and 19 from data obtained with the Vaisala RS80 and RS92 sondes. In particular, for $N_{max} \gg N_0$, the uncertainties rule the error and the sampling component may be discarded. For $N_{max} < N_0$ the sampling component dominates and the final error can be obtained directly from the $\epsilon_s$ model (Eq. 14).

The model was validated by comparison of poorly sampled profiles (only standard levels) and high resolution data. The result showed that the errors estimated for the low resolution profiles contain the high resolution values, considered the reference. The errors grew up to $> 30\%$ with poorly sampled profiles for dry atmospheres.

The median error is $2.4 \pm 0.9\%$, with ninetieth percentile $P_{90} = 5.0\%$ and an extreme value of $7.9\%$ for the driest condition (PWV$= 0.89\,\mathrm{mm}$). These results reduce in more than a half the uncertainties previously reported in the literature.

Therefore, not only the uncertainties are going to define the error in PWV estimations from radiosoundings, but the sampling is also playing an important role. Here we have proposed that it is possible to optimize the number of sampled levels to minimize the error within the instrumental



uncertainty. Whereas a radiosounding samples at least $N_0$ uniform vertical levels, that is depending on the dryness of the atmosphere, the error in the PWV estimation is likely to stay below $\approx 3\%$ ($P_{75} = 4.1\%$) even for dry conditions. Considering more samples than $N_0$ increase the noise and no the the information and, therefore, the accuracy in the estimation.

**Appendix A: Data compilation**

The following tables show the PWV and associated errors for each sounding at TFE and ORM.

*Acknowledgements.* This work has been funded by the Instituto de Astrofísica de Canarias (IAC). The radiosonde balloons were launched at ORM by the IAC and the University of Nice with support from the Centre National de Recherches Météorologiques (Météo-France). Soundings at Güímar are launched by the Spanish
Agencia Estatal de Meteorología (AEMET). Special thanks are due to the AEMET members Sergio Rodríguez, Ricardo Sanz and Ernesto Barrera for their valuable discussions and suggestions. Low resolution radiosounding data are available through the page of the Dept. of Atmospheric Science at the University of Wyoming [8]. We finally would like to acknowledge the constructive suggestions of an anonymous referee that clearly contributed to improve the paper.

---

[8]http://weather.uwyo.edu/upperair/sounding.html



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


**Table 1.** Saturation vapour pressure coefficients and associated uncertainties (Flatau et al., 1992).

| $a_j$ | Coefficient | Uncertainty |
|---|---|---|
| $a_0$ | 6.11176750 | $4.44010270 \cdot 10^{-01}$ |
| $a_1$ | $4.43986062 \cdot 10^{-01}$ | $2.86175435 \cdot 10^{-02}$ |
| $a_2$ | $1.43053301 \cdot 10^{-02}$ | $7.95246610 \cdot 10^{-04}$ |
| $a_3$ | $2.65027242 \cdot 10^{-04}$ | $1.20785253 \cdot 10^{-05}$ |
| $a_4$ | $3.02246994 \cdot 10^{-06}$ | $1.01581498 \cdot 10^{-07}$ |
| $a_5$ | $2.03886313 \cdot 10^{-08}$ | $3.84142063 \cdot 10^{-10}$ |
| $a_6$ | $6.38780966 \cdot 10^{-11}$ | $6.69517837 \cdot 10^{-14}$ |

**Table 2.** Relative errors $\epsilon_{rel}$ statistics (%): main percentiles ($P_{xx}$), median (med), minimum (min), maximum (max) and dispersion (s). The dispersion has been estimated robustly by means of $1.4826 \times \mathrm{MAD}$, where MAD is the median absolute deviation and 1.4826 is the scale factor between MAD and the standard deviation for perfect gaussian distributions.

| min | $P_{10}$ | $P_{25}$ | med–s | med | med+s | $P_{75}$ | $P_{90}$ | max |
|---|---|---|---|---|---|---|---|---|
| 1.4% | 1.8% | 1.9% | 1.6% | 2.4% | 3.2% | 4.1% | 5.0% | 7.9% |

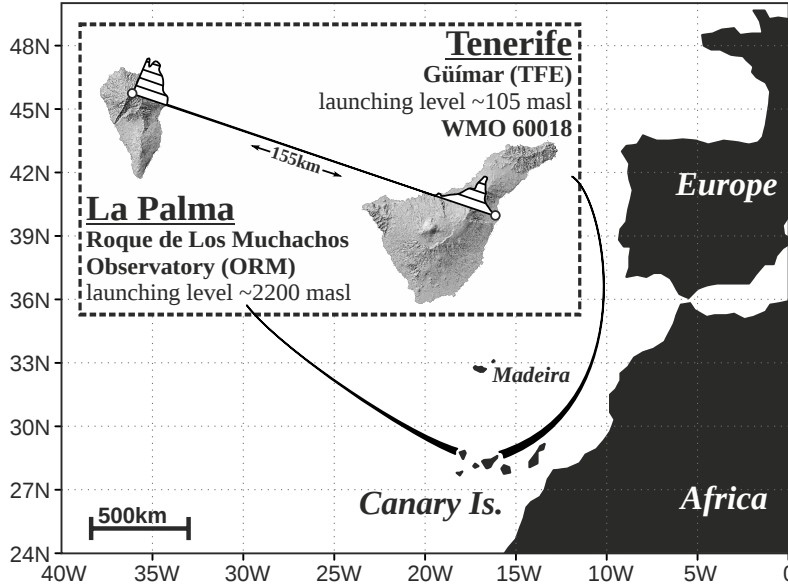

**Figure 1.** Location and characteristics of the two radiosounding launching points in the Canary Islands (ORM and TFE), separated by $\approx 155$ km, and the orographic profile linking the sites (the step in the altitude isolines of the orographic profile is 500m).





**Table 3.** PWV data and optimized error $\epsilon$ (see Eq. 15) for the 22 radiosoundings from ORM. $N_{max}$ is the actual number of sampled levels, $H_t$ (m) and $p_t$ (hPa) are the top height and pressure reached by the balloon, while $p_s$ (hPa) is the surface pressure at the launching point. The surface launching height is $\sim 2200$ m for all the flights. The parameters $\sigma_0$ and $\epsilon_{s0}$ are the uncertainty and sampling components of the error (Eqs. 17) and $N_0$ is the optimum number of levels (Eq. 16). The brackets in the last column are the relative errors $\epsilon_{rel}$.

| # | Ref | Date – Time (Y/M/D – UTC) | $N_{max}$ | $H_t$ (m) | $p_t$ (hPa) | $p_s$ (hPa) | $\sigma_0$ (mm) | $\epsilon_{s0}$ (mm) | $N_0$ | PWV (mm) | $\epsilon$ ($\epsilon_{rel}$) (mm; %) |
|---|---|---|---|---|---|---|---|---|---|---|---|
| 1 | VOL13 | 1990/04/02 – 02:03 | 3223 | 22685 | 37.4 | 790 | 0.06 | 0.04 | 806 | 0.89 | 0.07 (7.9) |
| 2 | VOL14 | 1990/04/02 – 05:45 | 3234 | 19757 | 60.1 | 789 | 0.08 | 0.07 | 462 | 1.84 | 0.11 (6.0) |
| 3 | VOL15 | 1990/04/03 – 00:08 | 3482 | 22455 | 38.7 | 790 | 0.15 | 0.12 | 268 | 4.53 | 0.19 (4.2) |
| 4 | VOL16 | 1990/04/03 – 03:03 | 3625 | 23284 | 33.7 | 788 | 0.15 | 0.12 | 259 | 4.64 | 0.19 (4.1) |
| 5 | VOL17 | 1990/04/05 – 04:26 | 2210 | 21429 | 45.6 | 782 | 0.21 | 0.16 | 201 | 6.29 | 0.26 (4.1) |
| 6 | VOL18 | 1990/07/13 – 22:00 | 2657 | 17797 | 83.7 | 793 | 0.24 | 0.18 | 178 | 7.94 | 0.30 (3.8) |
| 7 | VOL19 | 1990/07/17 – 02:22 | 7002 | 25537 | 24.2 | 792 | 0.25 | 0.18 | 171 | 9.68 | 0.31 (3.2) |
| 8 | VOL20 | 1990/07/18 – 00:20 | 4549 | 22648 | 37.9 | 790 | 0.13 | 0.10 | 304 | 3.25 | 0.16 (4.9) |
| 9 | VOL21 | 1990/07/20 – 02:45 | 4077 | 22749 | 37.2 | 789 | 0.18 | 0.14 | 227 | 4.45 | 0.22 (4.9) |
| 10 | VOL22 | 1990/07/21 – 00:00 | 2512 | 19646 | 61.3 | 791 | 0.18 | 0.14 | 229 | 3.56 | 0.22 (6.2) |
| 11 | VOL23 | 1990/07/21 – 03:21 | 3482 | 20683 | 51.6 | 791 | 0.16 | 0.12 | 249 | 4.52 | 0.20 (4.4) |
| 12 | VOL24 | 1990/07/21 – 22:57 | 4672 | 22542 | 38.5 | 793 | 0.16 | 0.10 | 312 | 3.81 | 0.19 (5.0) |
| 13 | VOL25 | 1990/07/22 – 03:06 | 3325 | 25818 | 23.1 | 792 | 0.12 | 0.09 | 333 | 2.63 | 0.15 (5.7) |
| 14 | VOL26 | 1990/07/22 – 22:10 | 4879 | 23126 | 35.2 | 792 | 0.11 | 0.07 | 444 | 2.59 | 0.13 (5.0) |
| 15 | VOL27 | 1990/07/23 – 01:41 | 4729 | 23793 | 31.7 | 791 | 0.08 | 0.06 | 526 | 2.04 | 0.10 (4.9) |
| 16 | VOL92 | 1995/11/01 – 11:00 | 3398 | 27987 | 15.7 | 790 | 0.18 | 0.10 | 309 | 3.83 | 0.20 (5.2) |
| 17 | VOL93 | 1995/11/03 – 00:50 | 3466 | 29257 | 12.9 | 793 | 0.08 | 0.05 | 578 | 1.83 | 0.09 (4.9) |
| 18 | VOL94 | 1995/11/03 – 22:20 | 2876 | 28593 | 14.4 | 793 | 0.29 | 0.21 | 152 | 12.9 | 0.36 (2.8) |
| 19 | VOL95 | 1995/11/04 – 02:00 | 2968 | 22755 | 35.5 | 791 | 0.31 | 0.21 | 149 | 12.9 | 0.37 (2.9) |
| 20 | VOL97 | 1995/11/08 – 23:08 | 3933 | 29481 | 12.1 | 763 | 0.13 | 0.09 | 328 | 3.68 | 0.16 (4.3) |
| 21 | VOL98 | 1995/11/09 – 01:30 | 3705 | 27971 | 15.6 | 783 | 0.17 | 0.14 | 218 | 5.44 | 0.22 (4.0) |
| 22 | VOL99 | 1995/11/09 – 04:00 | 4116 | 30828 | 10.2 | 782 | 0.19 | 0.14 | 229 | 6.85 | 0.23 (3.4) |





**Table 4.** PWV data and optimized error $\epsilon$ (see Eq. 15) from the first 21 radiosoundings of 42 launched at TFE.
See the caption of Table 3 for description.

| # | Ref | Date – Time (Y/M/D – UTC) | $N_{max}$ | $H_t$ (m) | $p_t$ (hPa) | $p_s$ (hPa) | $\sigma_0$ (mm) | $\epsilon_{s0}$ (mm) | $N_0$ | PWV (mm) | $\epsilon$ ($\epsilon_{rel}$) (mm; %) |
|---|---|---|---|---|---|---|---|---|---|---|---|
| 1 | 13/007B | 2013/01/07 – 12:00 | 3040 | 31132 | 9.1 | 1014 | 0.24 | 0.15 | 203 | 9.90 | 0.29 (2.9) |
| 2 | 13/135B | 2013/05/15 – 12:00 | 2530 | 28793 | 14.1 | 1003 | 0.30 | 0.20 | 159 | 16.53 | 0.36 (2.2) |
| 3 | 13/136A | 2013/05/16 – 00:00 | 2611 | 29672 | 12.4 | 1003 | 0.34 | 0.24 | 131 | 20.25 | 0.41 (2.0) |
| 4 | 13/148B | 2013/05/28 – 12:00 | 2629 | 29995 | 11.9 | 1006 | 0.27 | 0.19 | 165 | 13.62 | 0.33 (2.4) |
| 5 | 13/149A | 2013/05/29 – 00:00 | 2866 | 30415 | 11.2 | 1009 | 0.30 | 0.22 | 144 | 17.06 | 0.37 (2.2) |
| 6 | 13/153B | 2013/06/02 – 12:00 | 2628 | 29319 | 13.1 | 1004 | 0.25 | 0.19 | 165 | 11.81 | 0.31 (2.6) |
| 7 | 13/154A | 2013/06/03 – 00:00 | 2333 | 25585 | 23.3 | 1004 | 0.31 | 0.23 | 138 | 15.32 | 0.38 (2.5) |
| 8 | 13/165B | 2013/06/14 – 12:00 | 2473 | 27297 | 18.0 | 1005 | 0.33 | 0.24 | 131 | 22.50 | 0.41 (1.8) |
| 9 | 13/166A | 2013/06/15 – 00:00 | 2657 | 28438 | 15.3 | 1004 | 0.23 | 0.14 | 222 | 10.84 | 0.27 (2.5) |
| 10 | 13/182B | 2013/07/01 – 12:00 | 2829 | 29560 | 12.9 | 1004 | 0.27 | 0.20 | 158 | 14.17 | 0.34 (2.4) |
| 11 | 13/183A | 2013/07/02 – 00:00 | 2908 | 31220 | 10.1 | 1005 | 0.30 | 0.24 | 133 | 22.61 | 0.39 (1.7) |
| 12 | 13/203B | 2013/07/22 – 12:00 | 2729 | 29786 | 12.5 | 1005 | 0.37 | 0.25 | 125 | 23.31 | 0.45 (1.9) |
| 13 | 13/204A | 2013/07/23 – 00:00 | 2744 | 30442 | 11.4 | 1005 | 0.35 | 0.25 | 125 | 22.45 | 0.43 (1.9) |
| 14 | 13/213B | 2013/08/01 – 12:00 | 2639 | 29736 | 12.6 | 1001 | 0.31 | 0.24 | 132 | 19.53 | 0.39 (2.0) |
| 15 | 13/214A | 2013/08/02 – 00:00 | 2557 | 25626 | 23.6 | 1004 | 0.35 | 0.24 | 135 | 23.15 | 0.42 (1.8) |
| 16 | 13/233B | 2013/08/21 – 12:00 | 3221 | 31950 | 9.0 | 1003 | 0.45 | 0.34 | 95 | 41.08 | 0.57 (1.4) |
| 17 | 13/249B | 2013/09/06 – 12:00 | 2229 | 27485 | 17.5 | 1003 | 0.36 | 0.30 | 107 | 26.60 | 0.47 (1.8) |
| 18 | 13/250A | 2013/09/07 – 00:00 | 2474 | 29406 | 13.1 | 1005 | 0.36 | 0.26 | 124 | 23.44 | 0.44 (1.9) |
| 19 | 13/273B | 2013/09/30 – 12:00 | 2638 | 29176 | 13.4 | 1005 | 0.47 | 0.27 | 120 | 29.30 | 0.54 (1.8) |
| 20 | 13/274A | 2013/10/01 – 00:00 | 2892 | 31355 | 9.7 | 1002 | 0.37 | 0.29 | 112 | 25.23 | 0.47 (1.9) |
| 21 | 13/288B | 2013/10/15 – 12:00 | 2731 | 29125 | 13.3 | 1008 | 0.34 | 0.22 | 144 | 17.24 | 0.40 (2.3) |



**Table 5.** PWV data and optimized error $\epsilon$ (see Eq. 15) from the last 21 radiosoundings of 42 launched at TFE.
See the caption of Table 3 for description.

| # | Ref | Date – Time (Y/M/D – UTC) | $N_{max}$ | $H_t$ (m) | $p_t$ (hPa) | $p_s$ (hPa) | $\sigma_0$ (mm) | $\epsilon_{s0}$ (mm) | $N_0$ | PWV (mm) | $\epsilon$ ($\epsilon_{rel}$) (mm; %) |
|---|---|---|---|---|---|---|---|---|---|---|---|
| 22 | 13/289A | 2013/10/16 – 00:00 | 2847 | 29637 | 12.3 | 1006 | 0.31 | 0.22 | 143 | 16.69 | 0.38 (2.3) |
| 23 | 13/329B | 2013/11/25 – 12:00 | 2647 | 27989 | 15.4 | 1001 | 0.32 | 0.25 | 127 | 20.40 | 0.41 (2.0) |
| 24 | 13/330A | 2013/11/26 – 00:00 | 2541 | 27461 | 16.6 | 1000 | 0.30 | 0.21 | 150 | 16.92 | 0.36 (2.1) |
| 25 | 13/334B | 2013/11/30 – 12:00 | 2433 | 28148 | 15.1 | 1001 | 0.40 | 0.29 | 111 | 32.85 | 0.50 (1.5) |
| 26 | 13/335A | 2013/12/01 – 00:00 | 2511 | 27390 | 17.0 | 999 | 0.44 | 0.31 | 105 | 35.15 | 0.54 (1.5) |
| 27 | 13/340B | 2013/12/06 – 12:00 | 2078 | 23451 | 31.2 | 1003 | 0.42 | 0.28 | 116 | 25.39 | 0.50 (2.0) |
| 28 | 13/341A | 2013/12/07 – 00:00 | 2453 | 26832 | 18.5 | 1007 | 0.40 | 0.30 | 107 | 28.81 | 0.50 (1.7) |
| 29 | 13/361B | 2013/12/27 – 12:00 | 2645 | 30235 | 11.3 | 1010 | 0.32 | 0.20 | 156 | 16.15 | 0.38 (2.4) |
| 30 | 13/362A | 2013/12/28 – 00:00 | 2656 | 28921 | 13.7 | 1012 | 0.25 | 0.20 | 157 | 13.86 | 0.32 (2.3) |
| 31 | 14/013B | 2014/01/13 – 12:00 | 2782 | 31080 | 9.9 | 1015 | 0.28 | 0.21 | 147 | 17.25 | 0.36 (2.1) |
| 32 | 14/014A | 2014/01/14 – 00:00 | 2612 | 30027 | 11.6 | 1011 | 0.29 | 0.18 | 175 | 16.40 | 0.34 (2.1) |
| 33 | 14/022B | 2014/01/22 – 12:00 | 2708 | 26329 | 20.0 | 1010 | 0.35 | 0.23 | 136 | 21.80 | 0.42 (1.9) |
| 34 | 14/023A | 2014/01/23 – 00:00 | 2766 | 30428 | 10.6 | 1010 | 0.26 | 0.18 | 173 | 12.49 | 0.32 (2.6) |
| 35 | 14/045B | 2014/02/14 – 12:00 | 2695 | 29279 | 13.0 | 1008 | 0.28 | 0.20 | 159 | 10.98 | 0.34 (3.1) |
| 36 | 14/046A | 2014/02/15 – 00:00 | 2495 | 28342 | 14.8 | 1006 | 0.22 | 0.14 | 227 | 9.34 | 0.26 (2.8) |
| 37 | 14/049B | 2014/02/18 – 12:00 | 2776 | 31016 | 9.8 | 1009 | 0.23 | 0.19 | 164 | 11.30 | 0.30 (2.7) |
| 38 | 14/050A | 2014/02/19 – 00:00 | 2448 | 26572 | 19.5 | 1013 | 0.28 | 0.19 | 164 | 14.20 | 0.34 (2.4) |
| 39 | 14/100B | 2014/04/10 – 12:00 | 3094 | 29384 | 12.4 | 1002 | 0.28 | 0.19 | 163 | 17.38 | 0.34 (2.0) |
| 40 | 14/101A | 2014/04/11 – 00:00 | 2610 | 29792 | 11.8 | 1002 | 0.34 | 0.25 | 125 | 23.02 | 0.42 (1.8) |
| 41 | 14/104B | 2014/04/14 – 12:00 | 2927 | 29719 | 12.0 | 1002 | 0.36 | 0.24 | 134 | 22.84 | 0.43 (1.9) |
| 42 | 14/105A | 2014/04/15 – 00:00 | 2823 | 30473 | 10.8 | 1004 | 0.35 | 0.24 | 135 | 19.78 | 0.42 (2.1) |



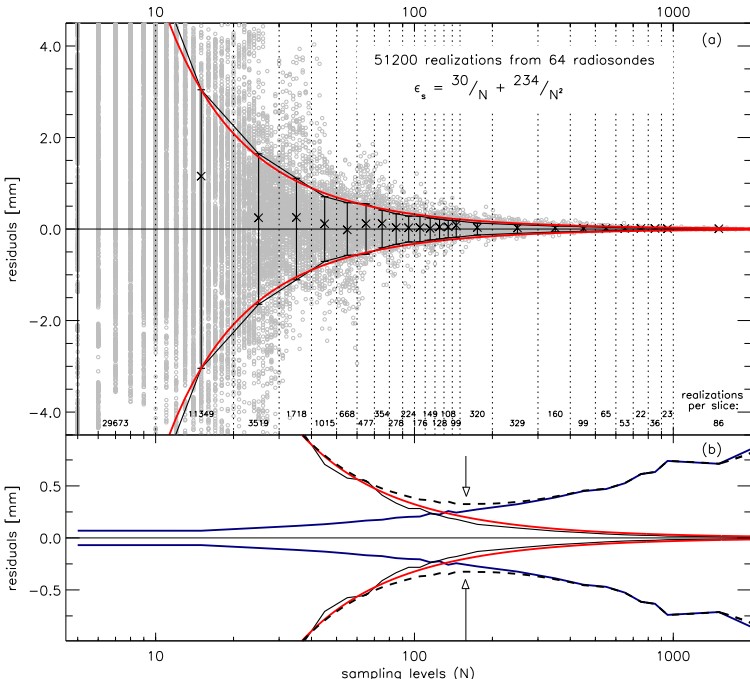

**Figure 2.** (a) Sampling error of radiosonde PWV as a function of the number of levels. The profiles were gradually sub-sampled to obtain 51200 realizations of PWV (grey circles). The error was fitted (red solid lines; see Eq. 14) to the RMSE of the residuals (see Eq. 12). The sampling levels were sliced (dotted lines). The error bars show the RMSE per slice and the "x" dots are the bias. (b) Sub-sampling error optimization. The solid black line is the experimental sample error as a function of the number of levels and the red line is the model (same as in panel a but with trimmed $y$ axis). The blue line is the median of the propagated uncertainties (Eq. 8) for all the profiles and the dashed line is the final error (Eq. 7). The arrows show the optimized number of samples for the minimum error.





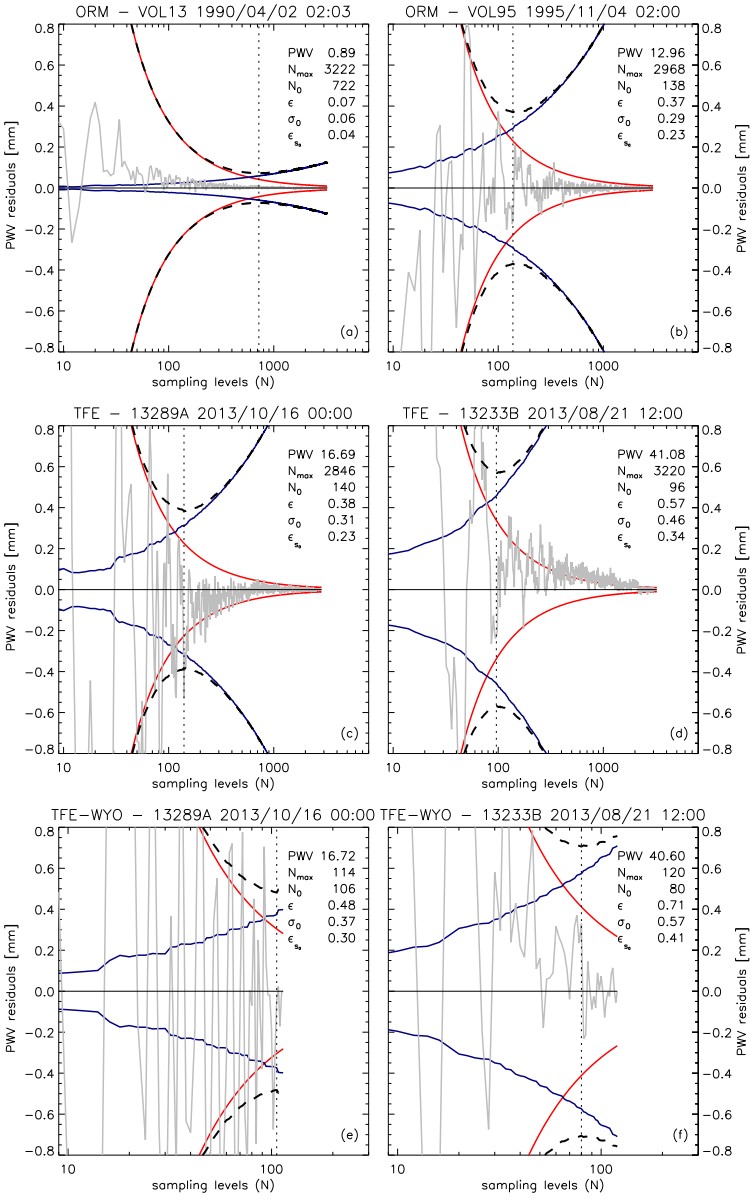

**Figure 3.** Six examples of sub-sampling error optimization for ORM (a) and (b) and TFE (c) and (d), with different PWV concentrations and $N_{max} \gg N_0$. Panels (e) and (f) show the same profiles that (c) and (d) but from data downloaded from the WYO repository (see section 1.1), where $N_{max} \approx N_0$. The solid grey line shows the residual between the sub-sampled PWV estimation and the best value with all the levels available. The vertical dotted lines are the optimized number of samples for the minimum error. The PWV and errors in the legend are in mm.





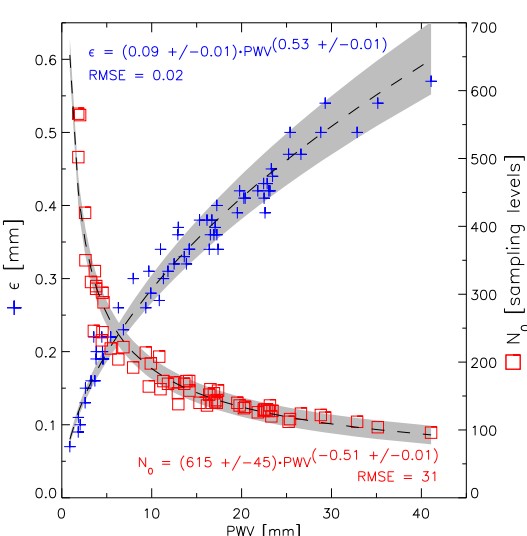

**Figure 4.** Optimized error $\epsilon$ (blue pluses and left axis) and optimum number of samples $N_0$ (red squares and right axis) for all the available radiosondes (ORM and TFE) as a function of the PWV value. The analytical equations are the best exponential fit to each data.



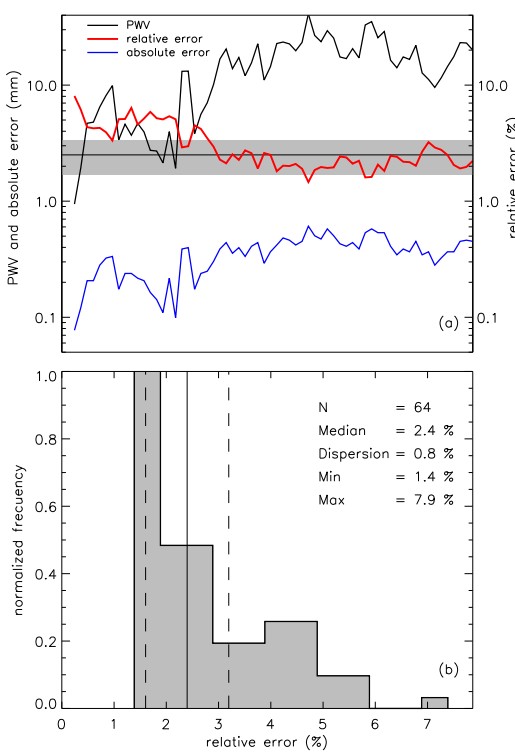

**Figure 5.** Relative error $\epsilon_{rel}$ statistics. Panel (a) shows all the PWV values (black), plotted together with their absolute (blue) and relative errors (red) in a logarithm scale. The thin horizontal line and shadow show the median relative error and dispersion. The dispersion has been estimated robustly by means of $1.4826 \times \mathrm{MAD}$, where MAD is the median absolute deviation and $1.4826$ is the scale factor between MAD and the standard deviation for perfect gaussian distributions. Panel (b) shows the histogram of the distribution of relative errors. The vertical lines show the median value and dispersion range (dashed). The main statistics is in the legend.



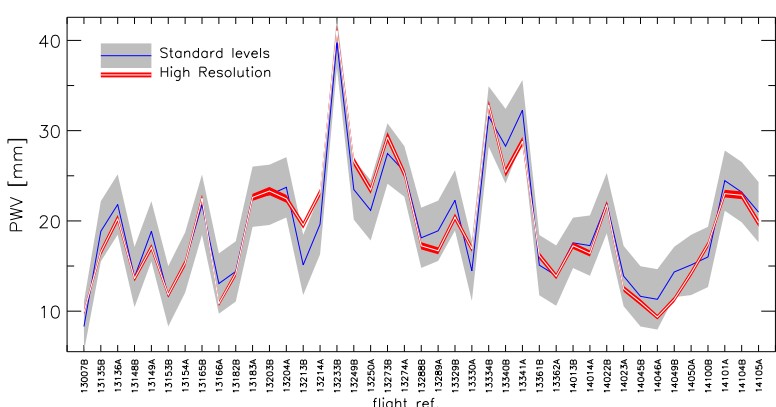

**Figure 6.** Comparison between the PWV series and errors (shadow) obtained from the TFE data with high resolution radiosoundings ($> 2500$ levels) and with the standard levels ($\sim 15$ levels). The colours are in the legend. The labels in the $x$ axis are the references for each balloon flight (see tables 4 and 5).