# Peer review of "A semi-empirical error estimation for PWV derived from atmospheric radiosonde data at the Canary Islands."

_Atmospheric Measurement Techniques, 2015_

## Referee Comment (RC1) · Anonymous Referee #2 · 7 Mar 2016

The paper investigates the error in Precipitable Water Vapor (PWV) derived from radiosonde profiles. Two error terms are considered: the measurement errors and the sampling error. It is found that the effect of the measurement errors increases with the number of levels sampled by the radiosonde, while the sampling error decreases with more levels sampled. Thus an optimum number of samples to be used in the PWV calculations is obtained. The subject is relatively interesting and the paper could be an good reference work for the accuracy of PWV from radiosondes. However, I have some doubt related to some parts of the results.

First of all, in principle you should expect that the more measurement points you include in the PWV calculations, the better the results ought to be, as long as an appropriate

method is used for the calculations. For example, I would expect that if the radiosonde measurements are smoothed to a lower resolution, the resulting PWV would be of higher accuracy compared to when reducing the resolution through sub-sampling. This ought to be discussed in the paper.

I have strong doubts relating the validity of equation (8), which is used for calculating the effect on PWV of the measurement errors. Applying the trapezoid method for calculating the PWV from eq. (3), we get:

$$PWV = \frac{10^5}{\rho g} \sum_{n=0}^{N-1} \frac{1}{2} \left(r_{i+1} + r_i\right) \left[p_{i+1} - p_i\right] \tag{1}$$

Thus the uncertainty of the contribution of the $i$:th layer to the PWV is (assuming $r$ and $p$ are uncorrelated) :

$$\sigma_i^2 = \frac{10^5}{\rho g} \left[ \frac{1}{4} \left(\sigma_{r_{i+1}}^2 + \sigma_{r_i}^2\right) \left[p_{i+1} - p_i\right]^2 + r_{i_avg}^2 \left(\sigma_{p_{i+1}}^2 + \sigma_{p_i}^2\right) \right] \tag{2}$$

By adding $\sigma_i^2$ of all levels, eq. (8) of the paper is obtained. However, this will only give the correct PWV uncertainty if the contributions from all layers can be considered independent. This is clearly not true, since the measurements at the top of one layer is also the ones used for the bottom of the next. Thus the layers are clearly not independent, thus eq (8) is incomplete. For a more proper derivation, we can use:

$$\sum_{i=0}^{N-1} (r_{i+1} + r_i)\Delta p_i = \sum_{i=1}^{N} r_i \Delta p_{i-1} + \sum_{i=0}^{N-1} r_i \Delta p_i \tag{3}$$

$$= \sum_{i=1}^{N-1} r_i(\Delta p_{i-1} + \Delta p_i) + r_0 \Delta p_0 + r_N \Delta p_{N-1} \tag{4}$$

Thus the contribution $\sigma^2_{mix}$ of the mixing ratio errors to the total uncertainty is:

$$\sigma^2_{mix} = \left(\frac{10^5}{2\rho g}\right)^2 \left[\sum_{i=1}^{N-1} \sigma^2_{r_i}(p_{i+1} - p_{i-1})^2 + \sigma^2_{r_0}\Delta p_0^2 + \sigma^2_{r_N}\Delta p_{N-1}^2\right] \tag{5}$$

Similarly the contribution from the pressure errors is:

$$\sigma^2_{pres} = \left(\frac{10^5}{\rho g}\right)^2 \left[\sum_{i=1}^{N-1} (r_{i-1,avg} - r_{i,avg})^2 \sigma^2_{p_i} - r_{0,avg}^2 \sigma^2_{p_0} + r_{N-1,avg}^2 \sigma^2_{p_N}\right] \tag{6}$$

And the total uncertainty is obtained by $\sigma^2 = \sigma^2_{mix} + \sigma^2_{pres}$, what differs significantly from the expression in eq. (8) in the paper.

The authors need to correct their expression for the uncertainty and redo all calculations.

Equation (9) in the paper ignores the contribution of the pressure measurement error to the error in the mixing ratio. Is this appropriate? Furthermore, in the denominator the approximation $p_i - e_i \approx p_i$ is made.

If the uncertainties of the saturation water vapor coefficients $a_j$ are taken into account when calculating the uncertainty of the partial pressure of water vapor (and hence also PWV), it should also be considered that the errors in these coefficients probably are the same for all measurements, thus these errors are highly correlated. Right now, the calculations assume that these error are uncorrelated.

Page 7, line 204: "Therefore, $\sigma$ increases with the number of levels in the profile": I do not necessary believe this is true. Of course, with each measurement you add some amount of error, however, on the other hand the contribution of each measurement to the total PWV error decreases.

I think the empirical expressions derived in eqs. (18) and (19) are only valid for the investigated location (Canary Islands). I would assume that, especially, the sampling

errors depend (at least slightly) on the location. Locations where, e.g., the humidity varies rapidly with height requires higher sampling than locations with more smooth variations.

---

## Referee Comment (RC2) · Anonymous Referee #1 · 22 Mar 2016

The work is of interest and potentially publishable in AMT. It is probably on the very brief end of what could be done and constitute a publication. I would hope that more in-depth analysis were feasible maybe using data from other locations which would serve to increase the impact and applicability of the findings. I would urge this to be pursued.

I also have a number of specific concerns regarding analysis experimental design that I detail in the major comments.

It would also be useful to know if and if so where processing software to perform the analyses is available from and under what licensing restrictions, if any.

[Figure]

Finally, the paper would also benefit from proof reading and language adjustments by a native English speaker prior to resubmission to make for an easier read.

Major comments

1. The RS-80 and RS-92 sondes are distinct models using (in the case of the humidity sensor very) distinct observing methods. It is probably unwise to consider them to be the same instrument type. Yet in several places this appears to be the case. It would make more sense to treat the RS92 and RS80 as distinct samples and at least show their equivalence.

2. Both sonde types have sensor response times that are likely lower than the archived measurement intervals for the high vertical resolution soundings. This is particularly acute for the humidity sensors in the upper-troposphere where the effective response time can extend to 30 seconds or more. This has the effect of smoothing the fields recorded vis-à-vis the true sampled state and reduces the effective degrees of freedom such that it is substantively lower than the implied profile measurement points count. Its not clear that this true data resolution, which is a function of the instrument performance rather than instrument measurement reporting frequency, has been adequately taken into account in Section 3 methodological approaches. Intuitively this would lead to an over-estimation of the effects of reduced sampling.

3. The reduced sampling 'standard' profiles are generally standard + significant levels (not just the levels in 5.1 which are too pessimistic an assumption) where the significant levels are defined as inflexion points in the T, RH or wind profile behaviour. This is a more information-rich sub-sampling than the options being considered in Sections 3.2, 4 and 5. It follows that the significant levels approach, which is akin to an optimal information content filter, will require fewer levels to recreate the salient profile features and total column estimates than those being considered in Section 5. The authors could get a trained operator to assign what significant levels would have been for the 64 high-resolution soundings and then repeat their analysis using standard and significant

levels. This would be a more useful and applicable comparison and increase the utility and value of their results. If they are already doing this in Section 5 early analysis this is not made sufficiently clear.

Minor comments

1. The final sentence of the abstract should be folded into the preceding paragraph rather than be a fragment sentence paragraph.

2. Line 98 change drooping to dropping

3. Lines 117-124 – while clearly these are reasonable estimates for the later RS-92 measurements based upon the referenced studies it is substantively less clear whether these assumptions hold for the earlier RS-80. In particular I'd expect on the measurement techniques a higher uncertainty on the humidity sensor for the RS-80 owing to use of a single sensor that could become contaminated.

---

## Author Comment (AC1) · 28 Jun 2016

We have carefully read and processed the suggestions made by the referees in two comments of the interactive discussion. As a result, we have deeply revised the manuscript and improved the results by introducing new data and analyses. Below is our response and actions taken for every referee comment:

**Report #1**
The referee urged us to extend the impact and applicability of our results by introducing data from other locations. We have followed the suggestion and looked for two other places with clear different atmospheric characteristics as Lindenberg

(LIN), in Germany and Ny-Ålesund (NYA), in the Svalbard Islands, Norway. We have also considered now separately the two locations at the Canary Is. (Roque de los Muchachos Observatory, ORM, at 2300 masl, and G"u'imar, GUI, at sea level). In summary, we have repeated the analyses at 4 different locations with equivalent dataset. The new data come from the GRUAN project. We have modified the Sec. 1.2 (Locations and datasets) and the map in Fig. 1 to include the details of the new sites. The title has also been slightly changed, removing 'Canary Islands', as the results includes now more sites.

Regarding the concern about the simultaneous use of data from the RS80 and RS92 sondes in the analyses. Some of the plots in the paper, as the Figs. 6 and 7, include all the data together. Nevertheless, now we have specifically indicated the range covered by each sonde type or location and we have not detected any inhomogeneity in the results. Any case, this was also a concern for us and one of the motivations for separating the locations at the Canary Is., as the RS80 data are only coming from the unique campaigns carried out in the ORM 20 years ago. In summary, now the RS80 and RS92 data are being treated as different samples, but no extra analyses have been carried out as we have not found inhomogeneities between them. We have included a new subsection (1.3 Instrumental setup) to discuss deeper this and other aspects of the sondes.

Another major comment by the referee was about how we have included the time lag delay of the RH sensors in our method. We have included a discussion of this issue in the Sec. 1.3 Instrumental setup. The sensor response time increases significantly in the upper troposphere for temp$< -68$C, that is reached typically above $10000$ m, where the water vapour mixing ratio drastically drops, reducing the contribution to the total integrated PWV. On the other hand, we can assume an uniform distribution in water vapour in the upper troposphere for the time delay. Under these conditions,

the time lag has the effect of smoothing the signal, reducing the dispersion, but not significantly moving the average and, therefore, not impacting in the total PWV in the column. This discussion has been included now in the paper.

From the point of view of the sub-sampling minimization process, we have proposed two different procedures to carry out the sub-sampling, one faster, taking one point in two, one in three, etc., and the other, by taking n, n-1, n-2, ... points per iteration, with more impact in the upper layers. We have found no differences between both approaches, unless for the very low resolution profiles.

The next major comment is noting that the use of only the standard levels to validate the model is a too pessimistic assumption. Actually, the original idea was to validate the estimated errors under the most unfavourable conditions. We agree with the referee that by using only the standard levels is not the most useful case and, following the suggestion, we have also processed the significant levels for the sample from GUI. Additionally, we have included the equivalent dataset from the repository of Wyoming. The new results are in a modified version of the last figure (Fig. 8) and in the discussion of Sec. 5.4, 'Validation for poorly sampled radiosonde data'.

The minor comments has been also taken in consideration. In particular, the last line of the abstract has been removed now as it has become obsolete after the new re-analyses. On the comment about the RH uncertainties on RS80. This is, in fact, the uncertainty documented by Vaisala. But, besides the discussion above on the homogeneity of the results from RS80 and RS92, we have also asked the team who released the balloons in that particular research campaigns at ORM (RS80) to confirm weather the calibrations and redundancy procedures were properly carried out. Any case, as previously suggested by the referee, we have now treated the sample with RS80 as a separate location (that actually is the less rich dataset) and, therefore, it

would be not impacting the method.

**Report #2**
The second referee showed strong doubts on the validity of our previous uncertainty propagation. This lead us to reconsider all our equations for the uncertainty component in the error and we think that this supposed an unique opportunity to improve the method. The main concern of the referee was on the validity of our approach for uncertainty propagation in the trapezoidal sum. We agree now with he/she and we have rewritten the final equations and going with more detail in an appendix section. In fact, our approach did not take into account what actually is a binning effect because of the double sum of each intermediate value, that are shared by two bins. This effect may also be read in terms of a full correlation between the upper part of one bin and the lower part of the next one. The new equations are now in the Sec. 3.1. We have followed the referee approach, that rearrange the summation to avoid the double contribution. In this way, the other binning correlations between adjacent levels, as the ones with saturation water vapour coefficients, are overcome in the final propagation.

We agree now with the referee in his/her doubts about the true of the assumption that the uncertainty increases with the number of sample levels. Using the new equations described above this is not true as, actually, the contribution of each measurement to the total PWV also decreases. But this is because we were considering only a variance component in the uncertainty. We have gone deeper with the correlation effects in the PWV summation and also considered the autocorrelation of the mixing ratio between each level and the incomings. We have discussed this in a new sub section (Sec. 3.2) and the modeled covariance component included in the new results. The results of the fitted exponential autocorrelation pressure lags obtained for each site are also considered in the discussion of the results, in the sub section 5.1, in terms of the WV distribution for each location.

[Figure]

One interesting consequence of the inclusion of the empirical covariances is that we have found a few outliers in the dataset of NYA, where the method found a limit. This particular points showed strong differences with the average WV vertical distribution in winter time, probably as a results of its latitude $\approx 80$N. For these cases the method must be specifically calibrated.

The referee pointed out another doubt on the approximation $p_i - e_i \approx p_i$. We have checked again this assumption and found differences less than 1e-6 in the final PWV content.

Finally, following a suggestion by the referee #1, we have now increased the potential impact of the method considering four different locations separately. In this sense, as pointed by the referee #2, the equations proposed for the Canary Is in the previous version have been rejected. Moreover, in the new results, there are no so clear correlation between the error and the total PWV.

**Final comments**
Following the referees points, we have corrected several parts of the manuscript that we detail for each section here:

*Title*
We have rejected Canary Is. now from the title, as we are presenting results for other sites.

*Abstract* (modified)

New results included and reference to the 4 sites now considered. Rejection of the reference to the particular equations for Canary Is. with the previous calculation. Some other minor corrections.

*1. Introduction* (modified)
Minor corrections in the first general introduction. Added reference to the new four sites in 1.1 'Rationale and objectives'. Added detailed info of the new sites in 1.2 'Locations and datasets'. Added new subsection with a more detailed discussion on the Instrumental setup (1.3)

*2. PWV from radiosondes* No changes.

*3. Error budget for PWV radiosonde data* (modified)
Inclusion of the covariance component. Correction of the variance component of the uncertainty propagation removing the binning correlation in sec. 3.1 'Uncertainty propagation. Variance component'. New sub section 3.2 'Uncertainty propagation. Covariance component'. The sampling error equation has been written in a general form with coefficients $a_0$ and $a_1$ in 3.3.

*4. Optimized error* No changes.

*5. Results* (modified)
Included a statistical summary of the PWV results. Separate sub sections for the discussion of the results of the autocorrelation exponential decay model and the sampling model for each site (5.1 and 5.2). Some changes in sub section 5.3 'Final errors'. Results updated. Previous equations specifically for Canary Is. rejected. NYA outliers discussion included. Added the reference to the other datasets with less levels
now included (Significant levels and Wyoming) in sub section 5.4 'Validation for poorly sampled radiosonde data'

*6. Conclusions* (modified)
Adaptations to the new results.

*Appendices* (modified)
Added the uncertainty propagation for the trapezoidal sum. Included the full data output tables from the 4 sites.

*Figures* (modified)
All the figures have been recalculated or adapted to the new dataset. New map. There are 2 new figures with the empirical parameters fitted for each location.

———————————————————